# Use Directional-Hemispherical Reflectance to Identify Female Skin Features in Response of Microdermabrasion Treatment

Anna Stolecka-Warzecha, Aleksandra Brandys, Małgorzata Bożek, Agata Lebiedowska *, Barbara Błońska-Fajfrowska and Sławomir Wilczyński

Department of Basic Biomedical Science, Faculty of Pharmaceutical Sciences in Sosnowiec, Medical University of Silesia, Kasztanowa Street 3, 41-200 Sosnowiec, Poland
* Correspondence: alebiedowska@sum.edu.pl; Tel.: +48-509923939

**Abstract:** Diamond microdermabrasion is one of the most popular cosmetological treatments in the field of mechanical, controlled abrasion of the stratum corneum corneocytes. In this study, the influence of micropeeling on the optical properties of the skin features was investigated. The directional reflectance of the skin was measured before and after the procedure. The study involved 15 women aged 23–26. The tests were performed on the surface of the skin in 15 spots—on the forearms, arms and lower legs. Post-treatment reflectance increased significantly in the wavelength range of 700–2500 nm. In the remaining ranges it also increased, except for the range 480–600 nm, in which it decreased, but insignificantly. It was found that the optical properties of the skin after microdermabrasion changed, the directional reflectance of the skin increased, especially in the infrared range. The main conclusion from the conducted research is that the directional-hemispherical reflectance method can be used to identify female skin features in response of microdermabrasion treatment.

**Keywords:** microdermabrasion; female skin; hyperspectral analysis; radiation reflectance

## 1. Introduction

Microdermabrasion is a mechanical and physical, controlled abrasion of the epidermis layers with a superficial effect with the use of appropriate equipment. Mainly the cells within the stratum corneum are exfoliated. The method is used in many dermatoses and skin defects. In order to achieve satisfactory and lasting results, it is necessary to perform a series of treatments [1]. Micropeeling is also used to increase the penetration of active ingredients of cosmetics, as the removal of lipophilic corneocytes facilitates the penetration of hydrophilic substances. The treatment can be performed on almost all skin types [2]. Microdermabrasion works in combination with other treatments, such as iontophoresis, sonophoresis, needle-free mesotherapy, chemical peels, and laser therapy [3]. The indications for the use of microdermabrasion are, among others: shallow wrinkles, scars, discoloration and skin laxity. Properly performed procedure has few side effects (reddening of the post-treatment area) and contraindications, which include bacterial, fungal, or viral infections at the site of the treatment, skin tumors and oncological therapy and acne vulgaris in the inflammatory stage [4]. The keratinized epidermis removal treatment can be performed using three methods that differ in the material used to exfoliate the epidermis and in the treatment intensity: diamond, corundum and water–oxygen microdermabrasion.

Exfoliation of hydrophobic corneocytes from the surface of S. corneum has a multidirectional effect on the skin and has an impact on the skin's features such as its hydration, sebum production, smoothness, elasticity, permeability and absorption of radiation. The depth of exfoliation has a direct impact on the subsequent effects of the treatment [5].

*Microdermabrasion and Directional Reflectance of Skin*

Electromagnetic radiation impinging on an object can be reflected, transmitted or absorbed. Which phenomenon will appear is determined by the wavelength and the

physicochemical properties of the body, such as the amount of chromophores in the skin—melanin or hemoglobin. The microdermabrasion treatment, i.e., the mechanical abrasion of the stratum corneum cells, affects the physicochemical properties of the skin, which can be measured using directional-hemispherical reflectance. The directional reflectance of a surface is defined as the ratio of the total energy reflected into the subtending hemisphere to the energy incident on the studied surface. The directional reflectance may be expressed in terms of primary quantities as:

$$\rho_d(\Theta_i, \phi_i) = \frac{\int_0^{2\pi} \int_0^{\pi/2} N_r \sin\Theta_r \cos\Theta_r d\Theta_r d\phi_r}{N_i \sin\Theta_i \cos\Theta_i d\Theta_i d\phi_i}$$

where:
$\rho_d$ is the reflectance;
$\Theta_i, \phi_i$ direction of energy incident on the surface;
$N_i$ is a radiance function of both position and direction, incident on the surface of an opaque object where some of the radiation is absorbed and the rest is reflected (includes diffuse reflectance or scattering)
$N_r$ is a radiance of the reflected radiation (also a function of position and direction) [6].

By measuring reflectance before and after the procedure, it can be observed how the skin parameters change, based on the data about how much radiation is reflected and how much is absorbed. The amount of energy absorbed by the skin provides information about the structure of its surface, as well as the number of chromophores it contains. The analysis of the measurement results provides information on whether there is a chromophore in the tested surface, and if so, what is its quantity. Based on the examination, the structure of the tested surface can also be assessed—if it is smooth, the radiation is absorbed more easily, but the more the structure is keratinized, granular or uneven, the easier it scatters the light. This research method can be used for assessing whether the procedure has been performed correctly and whether it is effective. The aim of the study was to analyze the effect of microdermabrasion on the optical properties of the skin using directional-hemispherical reflectance and to assess its impact on the skin's ability to absorb, reflect and scatter light. This aims to verify the usefulness of the directional-hemispherical reflectance method in assessing the effectiveness and the correctness of diamond microdermabrasion treatment.

## 2. Materials and Methods

### 2.1. Study Participants

The research was performed in accordance with the Declaration of Helsinki, on a group of 15 healthy women aged 23–26. Participants were characterized by Fitzpatrick skin phototype I–III. Fifteen spots were examined on the skin of the forearms, arms and lower legs. The inclusion criteria of participation in the study were good health, no visible dermatological changes and a at least six months break from performing other cosmetological procedures in the study areas.

Exclusion criteria from participation in the study were pregnancy, dermatological lesions such as moles and scars and any surgical procedures at the treatment area.

### 2.2. Procedure

Measurements were performed using a SOC 410 Solar DHR reflectometer from Surface Optics Corporation, San Diego, CA, USA. The reflectometer measures the integrated wave reflection coefficient—it is the ratio of the reflected energy to the total energy incident on the tested surface. It was measured for seven discreet wavelength bands, from ultraviolet, through visible light, to infrared: 335–380 nm, 400–540 nm, 480–600 nm, 590–720 nm, 700–1100 nm, 1000–1700 nm and 1700–2500 nm, at an angle of 60° and 20°. The part of the device that measures the reflectance was the hemispherical-directional head [6].

After cleansing the skin with micellar fluid, degreasing and disinfecting with a skin disinfectant, the hemispheric directional reflectance of each spot on the skin was measured

three times. The reflectometer was previously calibrated with calibration coupons. Then, the diamond microdermabrasion treatment was performed with a 150-grit head. After the treatment, the skin was disinfected and again the directional reflectance was measured three times.

## 3. Results

At a wavelength of 335–380 nm the reflectance increased after microdermabrasion in 66.7% of women (Figure 1). As a result, the median value (Me) and the first and third quartiles (Q1 and Q3) in the entire study group increased, respectively, Me: 0.119, Q1: 0.073, Q3: 0.156 to Me: 0.152, Q1: 0.090, Q3: 0.211 (Figure 2). However, these increases did not reach statistical significance.

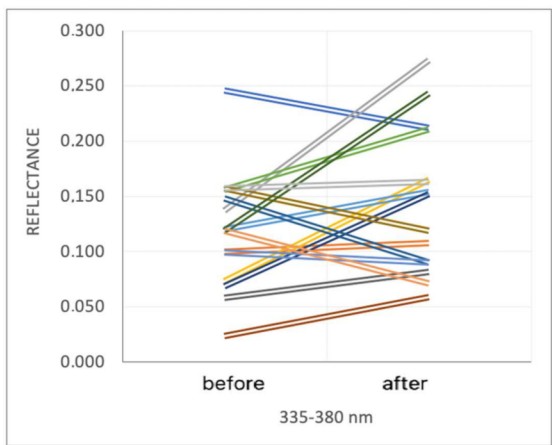

**Figure 1.** Average results of 3 reflectance measurements at a wavelength of 335–380 nm in every woman undergoing microdermabrasion, before and after the procedure.

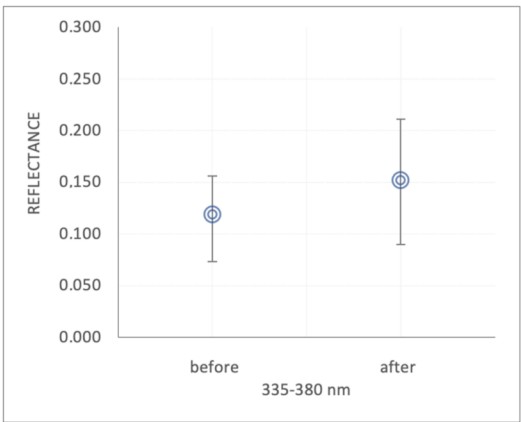

**Figure 2.** Median and quartile range of reflectance at the wavelength of 335–380 nm in the group of women undergoing microdermabrasion, before and after the procedure.

The increase in reflectance at a wavelength of 400–540 nm occurred only in 40.0% of women after microdermabrasion, in 6.7% the results did not change (Figure 3). Despite a slight increase in the median value from 0.345 before the procedure to 0.350 after the procedure, there was no significant effect of the microdermabrasion treatment on reflectance at the wavelength of 400–540 nm (Figure 4).

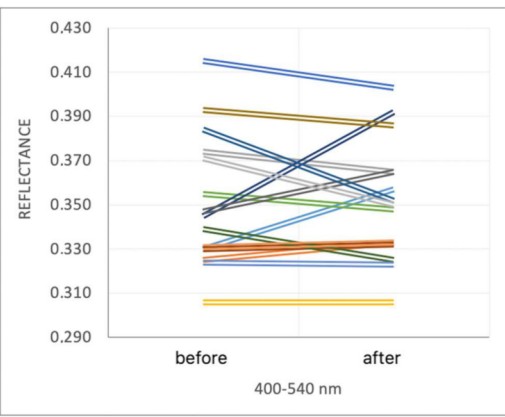

**Figure 3.** Average results of 3 reflectance measurements at a wavelength of 400–540 nm in every woman undergoing microdermabrasion, before and after the procedure.

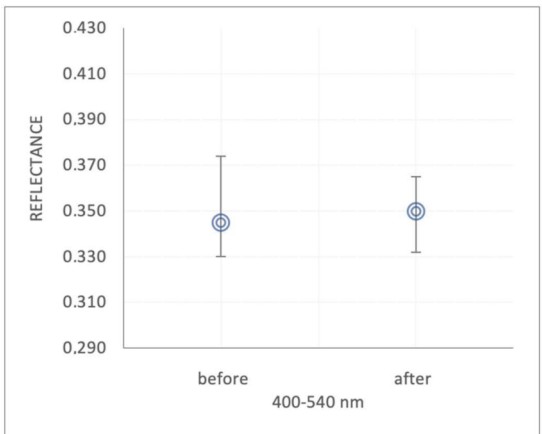

**Figure 4.** Median and quartile range of reflectance at the wavelength of 400–540 nm in the group of women undergoing microdermabrasion, before and after the procedure.

Microdermabrasion in 66.7% of women decreased the reflectance values at range 480–600 nm (Figure 5). The median, lower and upper quartile values before the procedure were, respectively, Me: 0.339; Q1: 0.334; Q3: 0.387, and after treatment, respectively, Me: 0.338; Q1: 0.321; Q3: 0.362 (Figure 6). The observed changes did not reach statistical significance.

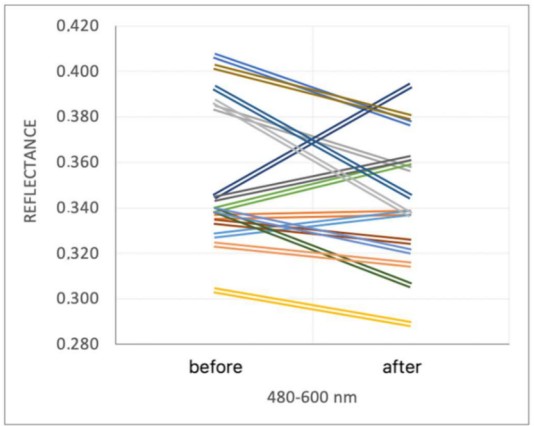

**Figure 5.** Average results of 3 reflectance measurements at a wavelength of 480–600 nm in every woman undergoing microdermabrasion, before and after the procedure.

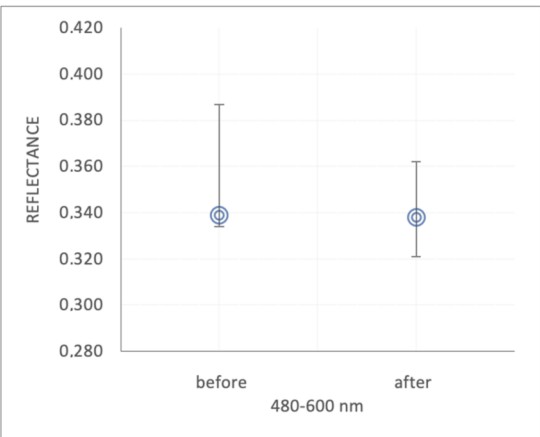

**Figure 6.** Median and quartile range of reflectance at the wavelength of 480–600 nm in the group of women undergoing microdermabrasion, before and after the procedure.

After the procedure, the reflectance at 590–720 nm increased among 73.3% of the examined women (Figure 7). The median and the quartile range increased after the procedure, but did not reach statistical significance. Before the procedure, the median and quartile range were, respectively, Me: 0.551; Q1: 0.534; Q3: 0.561; and after treatment, respectively, Me: 0.556; Q1: 0.535; Q3: 0.569 (Figure 8).

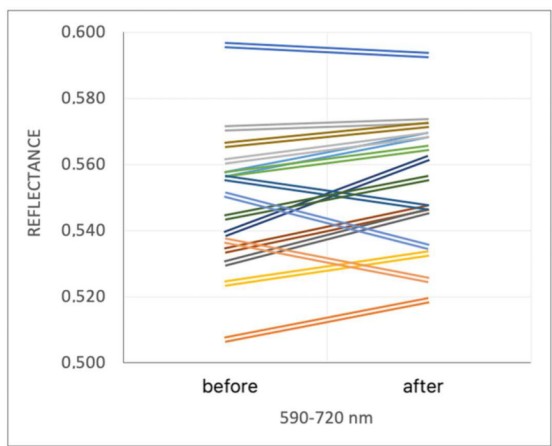

**Figure 7.** Average results of 3 reflectance measurements at a wavelength of 590–720 nm in every woman undergoing microdermabrasion, before and after the procedure.

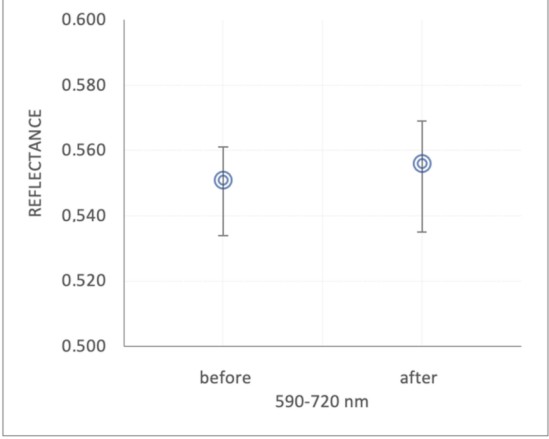

**Figure 8.** Median and quartile range of reflectance at the wavelength of 590–720 nm in the group of women undergoing microdermabrasion, before and after the procedure.

At a wavelength range 700–1100 nm, reflectance after microdermabrasion increased among 53.3% of women, and remained unchanged among 13.3% (Figure 9). Pre-procedure median and quartile range for reflectance were Me: 0.542; Q1: 0.531; Q3: 0.548, and after treatments Me: 0.542; Q1: 0.531; Q3: 0.548 (Figure 10). The increase in reflectance at a wavelength range of 700–1100 nm after microdermabrasion treatment reached statistical significance ($p = 0.017$).

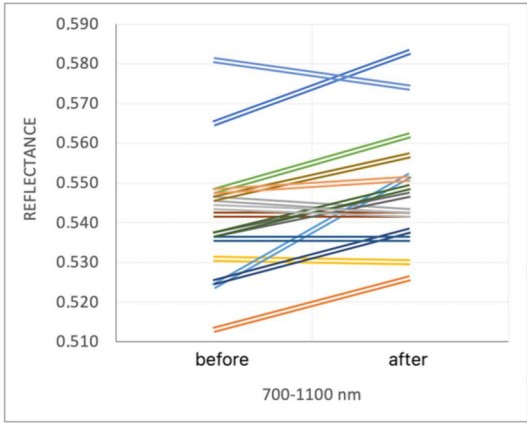

**Figure 9.** Average results of 3 reflectance measurements at a wavelength of 700–1100 nm in every woman undergoing microdermabrasion, before and after the procedure.

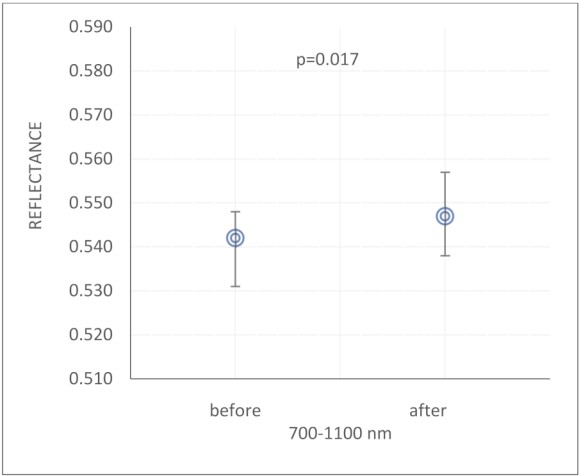

**Figure 10.** Median and quartile range of reflectance at the wavelength of 700–1100 nm in the group of women undergoing microdermabrasion, before and after the procedure.

Before microdermabrasion treatment, the reflectance at 1000–1700 nm was significantly lower than after the procedure ($p = 0.005$) (Figure 11) The increase in reflectance at 1000–1700 nm after microdermabrasion was observed among 80.0% of the examined women, and among 6.7% the measurement results remained unchanged (Figure 12). Before the procedure, the median reflectance was 0.280, the first quartile was 0.274 and the third quartile 0.289, after the procedure the median was 0.291, the first quartile was 0.286 and the third quartile was 0.310.

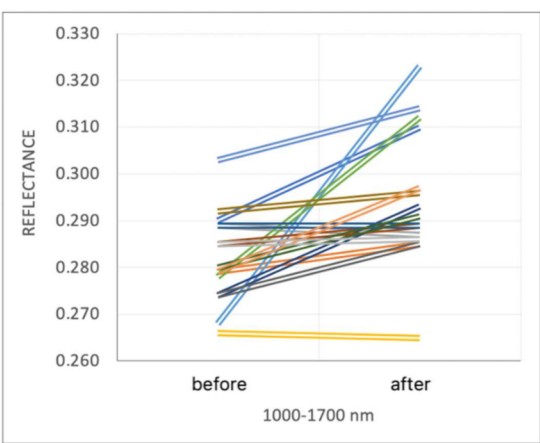

**Figure 11.** Average results of 3 reflectance measurements at a wavelength of 1000–1700 nm in every woman undergoing microdermabrasion, before and after the procedure.

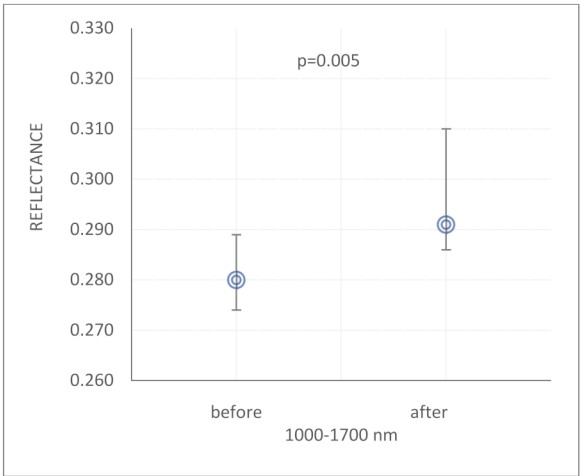

**Figure 12.** Median and quartile range of reflectance at the wavelength of 1000–1700 nm in the group of women undergoing microdermabrasion, before and after the procedure.

The largest statistically significant increase in reflectance after the microdermabrasion treatment occurred at the wavelength range1700–2500 nm and affected all the examined women ($p < 0.001$) (Figures 13 and 14). The pre-treatment median and quartile range of reflectance were Me: 0.280; Q1: 0.274; Q3: 0.289, and after treatments Me: 0.291; Q1: 0.286; Q3: 0.310.

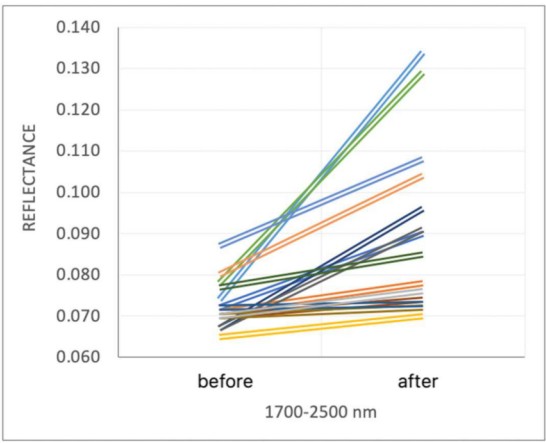

**Figure 13.** Average results of 3 reflectance measurements at a wavelength of 1700–2500 nm in every woman undergoing microdermabrasion, before and after the procedure.

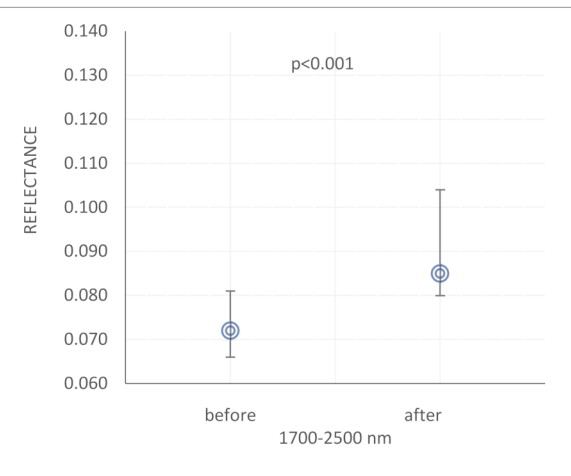

**Figure 14.** Median and quartile range of reflectance at the wavelength of 1700–2500 nm in the group of women undergoing microdermabrasion, before and after the procedure.

## 4. Discussion

The stratum corneum, as the subject of this research, and more precisely its structure and thickness, affects how the solar radiation hitting the skin is scattered/reflected. The correlation between microdermabrasion and directional reflectance of the skin indicates the effect of the treatment on increasing the reflection/scattering of electromagnetic radiation in the studied solar spectral range.

It should be emphasized that the impact of solar radiation on the skin largely depends on the wavelength. As the wavelength increases, the radiation energy decreases, and the depth of penetration of radiation into the skin increases. Studies on full-thickness skin fragments from the breasts of a 36-year-old woman (n = 3) have shown that, under normal conditions, UVA radiation (315–400 nm) can penetrate even into the dermis, while UVB (280–315 nm) and UVC radiation (100–280 nm) are absorbed almost entirely by s. corneum [7].

The microdermabrasion treatment has a double effect. First, reducing the thickness of the stratum corneum [8,9] causes the deeper penetration of the radiation, more absorption by skin chromophores and a decrease in reflectance. On the other hand, the microdermabrasion treatment immediately smooths the skin surface. A smooth surface reflects/diffuses radiation better than a surface with a high degree of roughness.

In 2016, Marczyk et al. [10] used corundum microdermabrasion, 50% lactobionic acid and the mixed method to demonstrate the effect of these treatments on reduction in sebum secretion. A similar study, but using a diamond microdermabrasion device on oily skin (Havemeister and Chilicka) [3], confirmed the effect of reducing sebum secretion, and also indicated the effect of increasing skin-hydration level, and reducing acne scars and open comedones. Hernandez-Perez and Ibiett [11] studied the effect of microdermabrasion on thick skin with dilated pores and a tendency to excessive sebum secretion. The results were also satisfactory considering the subjective opinion of patients, nurses and doctors. A biopsy, performed after a series of treatments, showed an improvement in the polarity of keratinocytes in the epidermis and a decrease in the fluidity of S. basale cells. Moreover, a decrease in the level of solar elastosis in the papillary layer and a slight reduction in the number of telangiectasias, inflammations and edema were noted [11].

After a 2–3 microdermabrasion sessions, it is possible to observe an increase in the number of ceramides in the epidermis, and thus improvement in its barrier function. However, it may be related to the previous damaging of the barrier as a result of S. corneum abrasion, which inducted the compensatory effect in the form of lipid level increase [12]. Microdermabrasion affects the thickening of the epidermis and connective tissue of the skin. Abdel-Motaleb et al. [13] also indicated an increase in the amount of collagen in the reticulate and papillary layers, which might reduce fine wrinkles on the skin surface.

After microdermabrasion treatments, a slight decrease in the amount of elastin fibers in the papillary and upper reticular layers was observed. However, Shim et al. [14] noted opposite results: an increase in elastin levels in two out of three cases, and no significant changes in collagen levels during the observations.

In other studies, in which one of the study groups consisted of people suffered from melasma, decreased melanization, and a more regular distribution of melanosomes in the epidermis were observed histopathologically [15]. In research by Shim et al. [14], the same changes were shown using the Fontan-Masson stain. However, the effect of the treatment on hyperpigmentation changes was questionable.

In our research, examining the skin before and after the micropeeling treatment, it was observed that the reduction in the thickness of the stratum corneum increased the reflectance coefficient in the range of the radiation length corresponding to UVA, UVB and UVC. This may be related to the alignment of the surface of the stratum corneum. This is an argument supporting the use of directional-hemispherical reflectance in assessing the correctness of micropeeling treatment, because the skin after the properly performed procedure will be characterized by higher reflectance. To study the effect of other procedures and active ingredients that are intended to smooth the skin surface, the suggested method can also be used. In the case of reflection/scattering of infrared radiation a particularly high correlation coefficient can be observed. It has been shown that in 100% of volunteers the increase in reflectance after the procedure occurred.

However, in the case of visible radiation with a spectral range of 480–600 nm, a decrease in the directional reflectance coefficient was observed after the micropeeling treatment. It may be related to increase in the absorption of radiation in this spectral range. Most of the skin's chromophores have absorption maxima in this spectral range. The absorption of oxyhemoglobin and deoxyhemoglobin peaks at wavelengths of 400–420 nm and 540–577 nm. In contrast, deoxyhemoglobin has an absorption maximum peak at 420 nm and a second peak at 580 nm. Oxidized hemoglobin (oxyhemoglobin) shows the highest absorption at 410 nm and 550–600 nm. Melanin also strongly absorbs radiation in this spectral range [16,17].

Therefore, it was suggested that the reduction in the thickness of the stratum corneum due to microdermabrasion treatment resulted in deeper penetration into the dermis by solar radiation beam (defined as the radiation energy range). The dermis and the deepest part of the epidermis—the basal layer—contain the vast majority of skin chromophores. The increase in the penetration of radiation into the dermis and the basal layer of the epidermis in the spectral range of 480–600 nm caused more efficient absorption of radiation by chromophores and the reflectance decreased.

It should be emphasized that the influence of microdermabrasion on the skin's direct reflectance coefficient for a wide spectrum of solar radiation has not been investigated yet.

The proposed method of directional-hemispherical reflectance can be used not only to assess the effectiveness of microdermabrasion treatment, but also to examine other procedures and products intended to affect the smoothness of the epidermis [18].

The decrease in the reflectance for the spectral range amount of 480–600 nm also indicated that more effective sun protection is needed after procedures that reduce the thickness of the stratum corneum. Radiation in this range carries less energy than ultraviolet radiation, but prolonged exposure and a relatively high penetration depth may have a clinically significant damaging potential. The implication of the results of Mahmoud et al. [19] was that visible light could possibly play a crucial role in producing darker and longer lasting pigmentation in populations with skin types IV–VI. Currently, there is no effective organic filter in sunscreens that protects against it. Moreover, a visible light has been shown to affect dermatoses such as melasma, post-inflammatory hyperpigmentation, solar urticaria, and erythropoietic protoporphyria [20]. However, there is a lack of cosmetic products that effectively protect against radiation other than ultraviolet radiation, because so far no standards or procedures have been developed to verify the effectiveness of sunscreen protection in the non-UV spectrum.

To sum up, the conducted research showed a significant influence of microdermabrasion on skin reflectance in the spectrum of solar radiation. An effective, quantitative method for measuring reflectance is the use of a directional-hemispherical reflectometer.

## 5. Conclusions

The main conclusion from conducted research is that the directional-hemispherical reflectance method can be used to identify female skin features in response of microdermabrasion treatment. Analyzing the measurements performed using a reflectometer, the additional conclusions can be drawn:

- The microdermabrasion treatment affects the optical properties of the skin.
- Microdermabrasion treatment causes a decrease in reflectance in the spectral range of 480–600 nm as a result of abrasion stratum corneum cells and, consequently, deeper penetration of radiation into the skin and absorption by skin chromophores.
- In the spectral range of 700–2500 nm, an increase in directional reflectance is observed after the microdermabrasion treatment, which may be related to the smoothing of the skin surface. Electromagnetic radiation is more effectively reflected from smooth surfaces than from rough ones.

Microdermabrasion treatment causing a decrease in skin reflectance in the spectral range of 480–600 nm requires additional sun protection. The key importance of this study is the conclusion that the directional-hemispherical reflectance method may be useful in assessing the effectiveness and correctness of microdermabrasion and other cosmetic procedures and treatments that affect the smoothness of the skin.

**Author Contributions:** Conceptualization, A.S.-W. and S.W.; methodology, A.S.-W. and A.B.; software, S.W.; validation, A.B. and A.L.; formal analysis, A.S.-W.; investigation, A.S.-W., A.B., M.B. and A.L.; resources, S.W. and B.B.-F.; data curation, A.S.-W.; writing—original draft preparation, A.L., A.B. and M.B.; writing—review and editing, A.S.-W. and S.W.; visualization, A.B. and M.B.; supervision, S.W. and B.B.-F.; project administration, A.S.-W. and A.B.; funding acquisition, S.W. All authors have read and agreed to the published version of the manuscript.

**Funding:** This research was funded by Medical University of Silesia, grant number PCN-1-166/N/1/0.

**Institutional Review Board Statement:** The study was conducted in accordance with the Declaration of Helsinki, and approved by the Bioethics Committee of the Medical University of Silesia (PCN/CBN/0022/KB1/27/III/16/17/21).

**Informed Consent Statement:** Informed consent was obtained from all subjects involved in the study.

**Data Availability Statement:** The datasets used and/or analyzed during the current study are available from the corresponding author on reasonable request.

**Conflicts of Interest:** The authors declare no conflict of interest.

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
