# Peer review of "Use Directional-Hemispherical Reflectance to Identify Female Skin Features in Response of Microdermabrasion Treatment"

_applsci, doi:10.3390/app122312087_

Round 1
Reviewer 1 Report
This work is interesting. However, the following comments should be considered:
1- In the Abstract, the main conclusion should be summarized, and it in the conclusion part should be rewritten;
2- The English should be improved.
Author Response
Anna Stolecka-Warzecha, Ph.D. Sosnowiec, 21st November 2022
Department of Basic Biomedical Science
Faculty of Pharmaceutical Sciences in Sosnowiec
Medical University of Silesia in Katowice, Poland
e-mail: astolecka@sum.edu.pl
Dear Reviewer,
I am very grateful for the valuable opinions and remarks. I agree with all the comments and I have referred to them as best as possible in a revised version of the manuscript. List of corrections is presented below:
This work is interesting. However, the following comments should be considered:
- In the Abstract, the main conclusion should be summarized, and it in the conclusion part should be rewritten;
Answer: The main conclusion has been summarized in the Abstract and rewritten in the Conclusions section.
- The English should be improved.
Answer: The English has been improved.

Reviewer 2 Report
There are 12 figures and 14 figure legends, please provide more information or include the figures which are missing. Other important point: is not possible to resume the number of figures?
please provide more information regarding the status Al treatment for these results.
please clarify the importance of this study.
Please include some references in the last paragraph from the discussion. Why is important to protect skin from visible light? How is possible to protect the skin?
Author Response
Anna Stolecka-Warzecha, Ph.D. Sosnowiec, 21st November 2022
Department of Basic Biomedical Science
Faculty of Pharmaceutical Sciences in Sosnowiec
Medical University of Silesia in Katowice, Poland
e-mail: astolecka@sum.edu.pl
Dear Reviewer,
I am very grateful for the valuable opinions and remarks. I agree with all the comments and I have referred to them as best as possible in a revised version of the manuscript. List of corrections is presented below:
- There are 12 figures and 14 figure legends, please provide more information or include the figures which are missing. Other important point: is not possible to resume the number of figures?
Answer: The two figures covered the other two, it has been corrected.
- please provide more information regarding the status Al treatment for these results.
Answer: The analysis of the results did not include artificial intelligence, due to the insufficient number of learning dataset. The number of volunteers was sufficient only to demonstrate statistical significance. And to use AI, data labelling is necessary. In order to effectively label the data, it is necessary to conduct research verifying the hypothesis about the possibility of using the proposed method and determine its sensitivity and specificity in relation to the analysed dataset first. And this research showed that. However, I will certainly use AI in the next stages of the research.
- please clarify the importance of this study.
Answer: The importance of this study has been clarified in Conclusion section.
- Please include some references in the last paragraph from the discussion. Why is important to protect skin from visible light? How is possible to protect the skin?
Answer: The paragraph about visible light and references to the paragraph have been added.

Reviewer 3 Report
Reviewer comments
The present article tilted “Use directional-hemispherical reflectance to identify female skin features in response of microdermabrasion treatment” via microdermabrasion treatment is important and the authors have shown the relevance of directional reflectance of the skin before and after treatment.
1. Authors need to provide more introduction.
2. In materials and methods could be written in more descriptive and clear manner.
3. In manuscript classifications section – Need to remove repeated words.
4. Overall, the manuscript can be accepted.
Author Response
Anna Stolecka-Warzecha, Ph.D. Sosnowiec, 21th November 2022
Department of Basic Biomedical Science
Faculty of Pharmaceutical Sciences in Sosnowiec
Medical University of Silesia in Katowice, Poland
e-mail: astolecka@sum.edu.pl
Dear Reviewer,
I am very grateful for the valuable opinions and remarks. I agree with all the comments and I have referred to them as best as possible in a revised version of the manuscript. List of corrections is presented below:
The present article tilted “Use directional-hemispherical reflectance to identify female skin features in response of microdermabrasion treatment” via microdermabrasion treatment is important and the authors have shown the relevance of directional reflectance of the skin before and after treatment.
- Authors need to provide more introduction.
Answer: More introduction has been provided.
- In materials and methods could be written in more descriptive and clear manner.
Answer: Materials and methods have been divided into sections and corrected to be more descriptive and clear.
- In manuscript classifications section – Need to remove repeated words.
Answer: We did not find repeated words. In which section of the manuscript? Can you specify?
- Overall, the manuscript can be accepted.

Reviewer 4 Report
In this manuscript, Lebiedowska and coworkers studied female skin features in response of microdermabrasion treatment with directional-hemispherical reflectance. This topic might be very interesting to many readers. The experimental results suggested that the microdermabrasion treatment could be evaluated by the optical properties of the skin, thus providing a general method to assess the correctness of the microdermabrasion treatment. Overall, this paper is well-written, logically reasonable, and probably novel to significant amount readers. Therefore, publication is suggested.
Author Response
Anna Stolecka-Warzecha, Ph.D. Sosnowiec, 21th November 2022
Department of Basic Biomedical Science
Faculty of Pharmaceutical Sciences in Sosnowiec
Medical University of Silesia in Katowice, Poland
e-mail: astolecka@sum.edu.pl
Dear Reviewer,
I am very grateful for the valuable opinions and remarks. I agree with all the comments and I have referred to them as best as possible in a revised version of the manuscript. List of corrections is presented below:
In this manuscript, Lebiedowska and coworkers studied female skin features in response of microdermabrasion treatment with directional-hemispherical reflectance. This topic might be very interesting to many readers. The experimental results suggested that the microdermabrasion treatment could be evaluated by the optical properties of the skin, thus providing a general method to assess the correctness of the microdermabrasion treatment. Overall, this paper is well-written, logically reasonable, and probably novel to significant amount readers. Therefore, publication is suggested.
Answer: Editing of English language and style has been performed.

Round 2
Reviewer 2 Report
Congratulations for this work